# Characterizing Surface Morphological and Chemical Properties of Commonly Used Orthopedic Implant Materials and Determining Their Clinical Significance

**DOI:** 10.3390/polym16091193

**Published:** 2024-04-24

**Authors:** Bertalan Jillek, Péter Szabó, Judit Kopniczky, Olga Krafcsik, István Szabó, Balázs Patczai, Kinga Turzó

**Affiliations:** 1Department of Orthopedics, Somogy County Mór Kaposi Teaching Hospital, Tallián Gyula u. 20-32, H-7400 Kaposvár, Hungary; 2Szentágothai Research Center, Environmental Analytical and Geoanalytical Research Group, Ifjúság útja 20., H-7624 Pécs, Hungary; sz.piiit01@gmail.com; 3Department of Optics and Quantum Electronics, University of Szeged, Dóm tér 9., H-6720 Szeged, Hungary; judit.kopniczky@gmail.com; 4Department of Atomic Physics, Budapest University of Technology and Economics, Budafoki út 8., H-1111 Budapest, Hungary; 5Department of Traumatology and Hand Surgery, University of Pécs, Ifjúság u. 13., H-7624 Pécs, Hungary; patczai.balazs@pte.hu; 6Dental School, Medical Faculty, University of Pécs, Tüzér u. 1, H-7623 Pécs, Hungary; turzo.kinga@pte.hu

**Keywords:** orthopedic implants, surface characterization, chemical composition, surface morphology, roughness

## Abstract

The goal of the study was to compare the surface characteristics of typical implant materials used in orthopedic surgery and traumatology, as these determine their successful biointegration. The morphological and chemical structure of Vortex plate anodized titanium from commercially pure (CP) Grade 2 Titanium (Ti2) is generally used in the following; non-cemented total hip replacement (THR) stem and cup Ti alloy (Ti6Al4V) with titanium plasma spray (TPS) coating; cemented THR stem Stainless steel (SS); total knee replacement (TKR) femoral component CoCrMo alloy (CoCr); cemented acetabular component from highly cross-linked ultrahigh molecular weight polyethylene (HXL); and cementless acetabular liner from ultrahigh molecular weight polyethylene (UHMWPE) (Sanatmetal, Ltd., Eger, Hungary) discs, all of which were examined. Visualization and elemental analysis were carried out by scanning electron microscopy (SEM), energy dispersive spectroscopy (EDS) and X-ray photoelectron spectroscopy (XPS). Surface roughness was determined by atomic force microscopy (AFM) and profilometry. TPS Ti presented the highest R_a_ value (25 ± 2 μm), followed by CoCr (535 ± 19 nm), Ti2 (227 ± 15 nm) and SS (170 ± 11 nm). The roughness measured in the HXL and UHMWPE surfaces was in the same range, 147 ± 13 nm and 144 ± 15 nm, respectively. EDS confirmed typical elements regarding the investigated prosthesis materials. XPS results supported the EDS results and revealed a high % of Ti^4+^ on Ti2 and TPS surfaces. The results indicate that the surfaces of prosthesis materials have significantly different features, and a detailed characterization is needed to successfully apply them in orthopedic surgery and traumatology.

## 1. Introduction

The biointegration or long-term functional stability of implant materials depends on several factors. The most important ones are the bulk and surface characteristics of the material, the biocompatibility and design of the material. Additionally, the applied surgical technique and the life quality or health awareness of the patient are essential issues [1].

Although the bulk properties (mechanical and thermal characteristics) of biomaterials are important with respect to their biointegration, the biological responses of the surrounding tissues to orthopedic implants are controlled mostly by their surface characteristics (chemistry and structure) because biorecognition takes place at the interface of the implant and host tissue.

Orthopedic implants are comprised of various materials, dependent upon the function they are intended to replace. Their biological behavior and survival can be controlled at the molecular and cellular level by the modification of the implant surface. Numerous modifications have been applied to medical implants in the past few decades to improve their functionality [2]. The optimal implant surface is different for any given purpose, thus, when the goal is to develop an implant surface, then the targeted functional part and the purpose of the modification has to be specified.

Many of these surfaces (and their modifications) are in the experimental stage and the in vitro, in vivo or clinical studies are still ahead. It is our belief that these characterizations will represent a huge positive contribution to clinical implant science and will help clinicians in selecting the optimal orthopedic implants for their patients.

Ideally, an orthopedic implant attaches rigidly to the bone during the patient’s remaining lifetime and must sustain strong forces whilst being pain-free. Orthopedic implant fixation can be classified as cemented fixation or biological (cementless) fixation. In cemented prostheses, bone cement poly-methyl-methacrylate (PMMA) is a grouting material positioned between the cement and the bone, resulting in two interfaces: implant–cement interface and cement–bone interface [3].

Long-term results of cementless joint replacement implants are largely determined by their integration into the receiving bone, known as osseointegration. Development of this biological process is essentially determined by the surface characteristics of the implanted device [4].

The most widely used metal for load-bearing implants is Titanium (Ti), due to its outstanding mechanical and biological properties. Titanium is a suitable implant material in every area of internal fixation. High corrosion resistance and chemical stability, excellent biocompatibility, bone apposition, lack of allergic reactions, low elastic modulus (high elastic flexibility), low weight and modifiable surface properties all sufficiently characterize Ti. In addition to these characteristics, reduced artifacts with magnetic resonance imaging (MRI) makes Ti a preferable choice for implant osteosynthesis [5]. Since its introduction as a medical implant in the 1950s, clinical demands have greatly increased. To meet these expectations, numerous modifications have been made regarding the alloy composition and the surface properties to achieve improved function and duration in the human body [6].

The surface roughness significantly determines the biointegration of a medical implant. In vitro studies and clinical experience proved that increasing the roughness of Ti alloy implants improves their osseointegration and osteogenic potential [7,8].

The currently used implant designs commonly seen in hip and knee replacements result in the formation of wearing particles at the articulating contact areas, which can lead to inflammation, osteolysis and inevitably, aseptic loosening of the prosthesis [9].

Its outstanding wear resistance and availability made Polyethylene (PE) a frequently used material in high-stress applications, such as total hip or total knee replacements. [10] Goswami and Alhassan developed a prediction model regarding UHMWPE in THR and TKR. The primary predicting factors were head diameter, body weight and head surface roughness (Ra) [11].

As an articulating surface, highly cross-linked UHMWPE revealed excellent results on wear rates in comparison with the conventional UHMWPEs. Cross-linking deteriorates the mechanical properties of UHMWPE limiting its utilization in high-stress contact applications such as TKA. However, when compared to UHMWPE, HXL showed 90% reduced wear rates [12,13].

Buford and Goswani compared wear rates of bearing material types. Titanium alloys and stainless steels resulted in increased wear rates when compared to ceramics and cobalt chromium alloys [14]. Stainless steels paired with polyethylene produced higher wear rates than when compared with cobalt chromium on polyethylene, and ceramic (Alumina) on polyethylene produced the lowest rates of the materials. Despite its good wear rates, the early ceramics had a high risk of fracture. Increasing its fracture resistance through altered manufacturing resulted in lower failure rates [15].

Zagra and Gallazzi reviewed bearing surfaces in primary total hip arthroplasty, in which they indicated that the application of the ceramic-on-ceramic or ceramic-on-polyethylene type bearing was dependent upon patient age and activity. Metal-on-polyethylene is still a valid option for older patients with good results lasting up to 15 years. Today, despite its early promising outcomes of hip resurfacing, metal-on-metal bearing is nearly entirely abandoned due to the adverse reactions caused by metal debris [16].

Total Knee Joint replacement (TKJR) prostheses consist of femoral, tibial and patellar components. The typical material regarding the femoral component is CoCr. Tibia components can be divided into monolith UHMWPE components or modular tibia components. Modular tibia components consist of a titanium alloy tibia tray into which the polyethylene can be inserted. The patellar component is made of UHMWPE with added titanium alloy for cementless use [3].

The aim of the present work was to conduct a thorough analysis of the surface characteristics (morphology and composition) of the above-described prostheses materials since their biointegration and long-term survival primarily depend upon their surface features. Determining the significant differences between the surface characteristics of different implant types widely used in orthopedic surgery and traumatology will highlight the importance of the appropriate knowledge of clinicians and adequate implant material choice. To the best of our knowledge, there has not yet been a study comparing these implant materials in this respect.

Material surfaces were visualized using Scanning Electron Microscopy (SEM), Atomic Force Microscopy (AFM) and profilometry. SEM created high resolution images for accurate imaging; furthermore, AFM and profilometry provided topographies and surface roughness values. Combined with the electron microscopic examinations, Energy-dispersive X-ray Spectroscopy (EDS) and X-ray photoelectron spectroscopy (XPS) were utilized for surface chemical characterization.

## 2. Materials and Methods

### 2.1. Sample Preparation

Discs of 1.5 mm thickness and 9 mm in diameter were fabricated from six different materials: (1) vortex plate anodized CP Grade 2 Titanium (Ti2); (2) non-cemented THR stem and cup CP Grade 5 Titanium alloy (Ti6Al4V) with titanium plasma spray coating (TPS); (3) cemented THR stem high nitrogen REX steel (SS); (4) TKR femoral component CoCrMo alloy (CoCr); (5) cemented cup from highly cross-linked UHMWPE (HXL); and (6) non-cemented acetabular liner from UHMWPE (Sanatmetal, Ltd., Eger, Hungary). The steps of the anodization were as follows: samples were degreased using SLOTOCLEAN AK 161 solution (Dr. Ing. Max Schlötter GmbH & Co. KG, Geislingen an der Steige, Germany) for 10 min, then rinsed off with deionized water, followed by immersion into a pickling solution containing hydrogen peroxide and hydrogen fluoride for 1 min. Following immersion, rinsing with deionized water and then cleaning using an ultrasound cleaner was performed for 80 s. Anodization for 180 s at 130 V and 1.8 A was performed, then samples were rinsed again for 30 s and dried for 10 min at 110 °C with compressed air. TPS discs were developed using a standard plasma spraying method. The surfaces of the samples were the same as in the case of prostheses used in orthopedic surgery. A standard sterilization method was applied in all samples.

### 2.2. Surface Characterization Techniques

Surface topography was visualized via SEM, AFM and profilometry. Elemental composition of the surfaces was measured using EDS and XPS. Roughness (R_a_) was assessed by AFM and profilometry.

#### 2.2.1. SEM-EDS

Samples were analyzed using a Jeol JSM-IT500HR (Jeol, Tokyo, Japan) scanning electron microscope (SEM) equipped with an integrated energy dispersive X-ray spectrometer (EDS). The use of the dry silicon drift detector (SDD) EDS enables rapid and highly accurate elemental analysis [17,18]. SEM is a reliable method for investigating metal surfaces. It can provide information on surface morphology, porosity and heterogeneity. In combination with EDS, chemical characterization can be performed in 1 μm depth of the sample. Samples were coated with gold (Jeol JFC-1300 auto fine coater, Jeol, Tokyo, Japan), and images were acquired in secondary electron imaging mode at 5, 10, and 15 kV accelerating voltage. The following magnifications were used: 100×, 250×, 500×, 1000×, 2500×, 10,000×. In some cases, the discs were tilted 45° for improved visualization.

#### 2.2.2. AFM

In consideration of AFM measurements, a PSIA XE-100 instrument (PSIA Inc., Seoul, Republic of Korea) was used to study the surface morphology and roughness (R_a_) of the samples. AFM is a high-resolution imaging technique used to study surfaces in the μm to nm range by measuring the forces acting upon the AFM probe tip as it approaches and retracts from the surface under study. The tips were single-crystal silicon cantilevers (type: N, NSG30 series with Au reflective layer, resonance frequency 240–440 kHz, force constant 22–100 N/m) from NT-MDT (Moscow, Russia). Measurements were performed in tapping mode, taking height, deflection and 3D images of 5 µm × 5 µm, 20 µm × 20 µm and 40 µm × 40 µm area. R_a_ was determined as the arithmetic mean of the surface height versus the mean height using XEI 1.6 (PSIA Inc.) AFM image processing program (with a minimum of ten independent measurements).

#### 2.2.3. Profilometry

For profilometry measurements, a Veeco, Dektak 8 Advanced Development Profiler^®^ (Veeco Instruments, Plainview, NY, USA) was used. The tips featured a radius of curvature of ∼2.5 µm and the force applied to the surface during scanning was ∼30 µN. The imaging resolutions in the x (fast) and y (slow) scan directions were 0.33 µm and 9.52 µm, respectively. The vertical resolution was 40 Å. Surface topography of 500 × 500 µm^2^ and 1000 × 1000 µm^2^ was recorded on each sample, and average roughness values (R_a_) were calculated using Vision^®^ for Dektak, Version 3.42 (Veeco Instruments Inc.) imaging software (with reference to the ANSI B46.1 surface texture specification).

#### 2.2.4. XPS

Additionally, the surface composition of the samples was analyzed by XPS using a twin anode X-ray source (XR4, Thermo Fisher Scientific, Waltham, MA, USA) and a hemispherical energy analyzer with a 9-channel multi-channeltron detector (Phoibos 150 MCD, SPECS). The base pressure in the analysis chamber was approximately 2 × 10^−9^ mbar. Ti2, TPS, SS, CoCr, HXL and UHMWPE samples were analyzed using a Mg Kα (1253.6 eV) anode without monochromatization. Peak fitting was performed using CasaXPS software version 2.3.26. Wide-range scans and high-resolution narrow scans were performed for all samples. XPS carried out the identification of the atoms on the top layer of the surface in the depth of 1–10 nm.

### 2.3. Statistical Analyses

Arithmetic means ± the standard error of the mean (SEM) were calculated for R_a_ (nm) values measured by AFM. Following normality testing, the data were compared using one-way analysis of variance (ANOVA), followed by Tukey’s HSD, LSD, and Scheffé post hoc tests to detect statistical differences after multiple comparisons (SPSS 21, SPSS, Chicago, IL, USA). The significance level was set at *p* = 0.05.

## 3. Results

### 3.1. SEM

SEM micrographs depicted typical surface topographies and structures for the different sample discs (Figure 1, Figure 2 and Figure 3).

Smaller magnifications (100–500×) visualized the coarse superficial structures. Ti2 alloy discs showed regular, 100 μm wide circular grooves loosely perforated with small openings, while the TPS-coated sample exhibited an irregular, much rougher picture with round structures in various sizes, nearly twenty to a few hundred μm in width (Figure 1a,d).

When increasing the magnification (1000–2500×), SEM revealed a more detailed picture of the surfaces. The Ti surface had circular and oblong, 1–5 μm wide, round-edged cavities occurring frequently (Figure 1b). The TPS surface was rougher with variable-sized overlapping droplets made by ”splashing” during the plasma spraying mechanism. These structures are laid out on a wave-like flat layering surface (Figure 1e).

The highest magnification (10,000×) recordings provided increasingly enhanced representation of the submicron formations and their dimensions. Small cavities of the Ti sample were nearly 0.5 to 3 μm, round apertures with a seemingly more extended diameter under the surface with additional small holes appearing on the bottom (Figure 1c). The spherical droplets of the TPS surface were covered with 4–5 μm wide similar round structures mostly covering their surfaces (Figure 1f).

SS samples had a smooth surface with wave-like folds, nearly 100 μm in length (Figure 2). At a higher magnification, various-shaped tiny particles and short scratches in an irregular pattern were seen on a smooth surface. We observed these small granules which are randomly scattered over the SS surface, and they are diverse in size and shape (Figure 2a,b). The mostly round and sharp-edged crystal-shaped particles were sized between tenth of a μm to a few μm in length (Figure 2c).

The investigated CoCr discs showed a dense grainy pattern with numerous small parts marginated by sharp edges (Figure 2). At higher magnifications, CoCr showed a much more irregular picture with 10–20 μm long, sharp-edged scratches in randomized directions (Figure 2d). Scratches covering the CoCr samples are densely cracked with sharp elevations creating a harsh surface with crevices (Figure 2e,f).

HXL and UHMWPE had a similar picture at 100× magnification with narrow circular creases consisting of densely spaced narrow parallel lines (Figure 3a,b,d,e). At higher magnifications, UHMWPE shows 10 μm long narrow scratches perpendicular to the creases, yet these scratches were not observed on HXL.

At 10,000× magnification, SEM images revealed HXL had many particles with a few hundred nm in size, while UHMWPE showed shallow scratches perpendicular to the creases. These scratches are more defined and regular, in which small particles are not visible.

### 3.2. AFM

We performed AFM examinations on the Ti2, SS, CoCr, HXL and UHMWPE samples. The 5 μm × 5 μm, 20 μm × 20 μm and 40 μm × 40 μm size areas were scanned. TPS samples had a roughness over 10 μm, therefore a profilometric study was completed instead of AFM.

AFM and profilometry examination provided additional visualization of the surface morphology and the determination of roughness (*R_a_* (nm)) of the investigated implant materials.

Pictures revealed by AFM correlated with the images made by SEM, depending on the area of scanning. The 40 × 40 μm topographies corresponded to the smaller magnification SEM results and showed the same general patterns of the samples (Figure 4). Scanning of smaller areas resulted in a similar sight to the highest magnifications used at SEM measurements.

At the 40 × 40 μm scan size, HXL and UHMWPE featured the smoothest surface with an average roughness values (R_a_) of 147 ± 13 nm and 144 ± 15 nm, respectively (Figure 4 and Figure 5). SS had a somewhat rougher surface of 170 ± 11 nm, followed by Ti2 with 227 ± 15 nm. CoCr proved to have the roughest surface of the samples investigated by AFM with a R_a_ value of 535 ± 19 nm.

The average roughness values measured at different scanned size areas were directly proportional to the field of view (FOV); however, their relative difference showed an inverse proportionality compared to the investigated area. The biggest relative difference between the mean R_a_ values was detected evaluating the 5 × 5 μm area. Increasing the FOV (20 × 20 μm and 40 × 40 μm) resulted in an increase in the measured mean values; however, their relative difference decreased.

### 3.3. Profilometry

All materials were scanned by the profilometer with areas of 500 μm × 500 μm and 1000 μm × 1000 μm (Figure 6 and Figure 7). This method provided information nearly 25× larger area than the AFM; therefore, a bigger surface of the sample was characterized. These results with respect to topographies were comparable with lower magnification SEM pictures, yet the roughness values were different from the AFM measurements due to the dissimilar FOV. 

The roughness of the Ti2 was minimally higher than the polyethylene surfaces (Ra: 1.7 ± 0.1 μm) (Figure 8). Recurrent grooves of Ti2 surfaces were also visualized using this method. The heights of these grooves were 4–8 μm, measured from the base (Figure 6A).

In accordance with its outstanding roughness (R_a_: 25.5 ± 2.8 μm), the TPS surfaces consisted of structures in various height and width in an irregular, dense arrangement (Figure 6B and Figure 8).

SS samples had an average roughness of 2.5 ± 0.1 μm and their topography showed wave-like patterns with 2–9 μm high peaks above the mean line (Rp) (Figure 6C and Figure 8).

CoCr had the least rough surface with profilometry, as R_a_ was 1.0 ± 0.1 μm. (Figure 6D and Figure 8). The topographies showed CoCr had an irregular, granular surface, in accordance with the 100× magnification SEM pictures.

HXL and UHMWPE surfaces looked very similar when measured using profilometry and had a relatively smooth surface (1.4 ± 0.1 μm and 1.5 ± 0.1 μm, respectively). It showed the same circularly creased topography as seen at low magnification SEM images (Figure 7 and Figure 8).

### 3.4. EDS

Energy Dispersive Spectroscopy provided elemental analysis of the surfaces. Ti2 consisted mostly of O, Ti and a small amount of C, with a minimal proportion of P. TPS also consisted mainly of O and Ti. TPS contained twice as much titanium as Ti2; however, it showed lower amounts of C and higher amounts of N and Al with a small presence of Si and Na (Table 1).

More than half of the SS surface consisted of Fe (Figure 9), and about half of the CoCr consisted of Co; however, the second most common element was Cr in both samples. Both contained approximately the same proportions of C. Additional elements showed varied amounts (Table 2).

Both HXLPE and UHMWPE surfaces consisted of C atoms with a small ratio of contaminations of Al in both samples and Ni in the case of HXL (Table 3).

### 3.5. XPS

XPS supported the results of EDS on elemental composition; furthermore, it provided information in reference to the binding state of the atoms. It revealed a high proportion of Ti^4+^ on Ti2 and TPS surfaces (Table 4 and Table 5).

Important differences were shown in the case of Ti2 and TPS samples. The Ti2 sample has an oxide layer which is more homogenous in all respects (thickness and bonding state surface) when compared to the TPS sample, in the very least, in the information depth of XPS (Table 5 and Table 6). On the other hand, the TPS sample shows a metallic Ti bonding state (Table 5, Figure 10). Plasma spraying can create peculiar surface bonds, and we assume this oxide layer is not as homogeneous in depth as when compared with the Ti2 (anodized) sample.

XPS showed additional elements on PE surfaces and detected the binding states of C and O atoms. Similar atomic composition was detected, with a small deviation in the detected elements. A small amount of Zn and Na was observed on the UHMWPE surface (Table 7).

Differences were revealed between the binding states of oxygen atoms of the PE materials (Table 8).

Dissimilar O binding states were detected on the PE surfaces, due to various manufacturing processes (cross-linking method) applied for these materials.

## 4. Discussion

SEM, AFM and profilometry studies proved significant differences and typical features of the surface morphology regarding our investigated materials. Profilometry showed different values when compared to AFM due to the different FOVs. EDS and XPS revealed typical elements on the investigated materials.

The Anodized Ti2 surface showed concentric grooves created by the turn mill during the processing of the sample. The granular morphologic structure overlapping the grooves is the result of the anodization process. AFM at the 40 μm × 40 μm areas revealed the overlapping structures, while smaller FOVs revealed the precise characteristics of the granules. Our previous study proved that the surface roughness determined by AFM is dependent upon the field of measurement due to the different macroscopic features of the surfaces [19].

Therefore, in consideration of a thorough characterization of biomaterials, it is advisable to measure roughness at different scan sizes. Profilometric determination of the roughness gives the characterization of a larger surface of the samples, hence, in case of complex surfaces, it is worthwhile to determine the roughness by both methods (AFM and profilometry).

In its role as bone plate material, titanium provides many advantages when compared with stainless steel. Ti alloys match the modulus of elasticity of the bone better, it provides increased strength, and it is more bioinert. Formation of a self-regenerating TiO_2_ layer on its surface provides corrosion resistance. However, removal may be difficult due to good osseointegration of the implant and possible cold welding between the screws and plate [20].

Anodization offers ideal bioactive surface properties for Titanium implants. It can provide a porous, rough surface with higher surface energy and ideal hydrophilic properties for osseointegration [21].

Kim et al. and Mühl et al. reported the formation of a rougher and thicker oxide layer as the result of anodization on commercially pure Ti used for orthopedic and dental implants, demonstrated by SEM-EDS, AFM and XPS [19,22]. Traini et al. find that anodized titanium surfaces have a high ability inducing fibrin formation thus accelerating osseointegration and it showed a significantly higher bone-implant contact rate compared to non-anodized implants. They also measured the nano-roughness of anodized dental implant surfaces in which they found a value close to the results of our investigation (Ra: 286 ± 40 nm) [23]. Yildiz et al. performed profilometry measurements on Ti discs with an anodized SLA (sand blasted, large grit, acid etched) surface used as dental implants and also produced similar results (Ra: 1.39 μm) [24].

Studies showed that anodized dental implants provide promising results with a low rate of marginal bone loss [25].

TPS discs revealed a much rougher, irregular surface, determined with profilometry and SEM-EDS. Titanium plasma spraying creates typical surfaces with random droplet-shaped scatters. Our SEM-EDS and XPS measurements revealed a high proportion of Ti^4+^ on Ti2 and TPS samples, which confirms the presence of a thick oxide layer on their surfaces. In addition to these findings, the anodized sample has a more homogenous and denser oxide layer when compared to the TPS sample. The altered topography generated by Titanium plasma spraying increases its tensile strength on the bone/implant interface, as reported by Buser et al. [26].

Studies showed that the thick oxide layers detected on TPS samples have an enhancing effect upon bone formation [27,28].

Titanium plasma spraying is a common method for increasing the surface in cementless hip prostheses. As mentioned above, increasing the roughness favors osseointegration, yet also facilitates bacterial adhesion [29].

Lombardi et al. followed up on 2000 tapered titanium-porous-plasma-sprayed THR cementless femoral components. They reported a 95.5% survival rate at 20 years with a low rate of septic revisions (0.4%) [30].

**Stainless steel** plates showed a much smoother surface in contrast to the TPS discs.

Wu et al. investigated the bacterial adhesion and microcolony formation on unpolished and differently electropolished stainless steel surfaces. The AFM-measured average roughness of the unpolished surface was in the same range as our R_a_ value. Their XPS measurements showed the unpolished surface contained similar proportions of C, O, Fe and N, much as in the case of our samples. On their surfaces, there was also a small amount of Cr; however, unlike ours, it did not contain any other elements such as Al or Ni [31].

Stainless steel was the first class of alloy introduced for orthopedic implants and it is still used as a cemented hip prosthesis material [32]. Studies demonstrated that, particularly for cemented THR designs, better results are gained with smooth surfaces as compared to rough surfaces [33]. PMMA is vulnerable to tensile stresses and shear forces, yet it tolerates compressive loads. A polished, smooth surface results in a weak cement–stem bond. Poorly bound, tapered stems do not create tensile and shear stresses in the cement and the cement–bone interface. In contrast, designs with a rough surface can result in detrimental debris formation causing the loss of bony support (osteolysis) of the implant and inevitably, the loosening of the implant [34].

In the case of cement fixation, the stainless steel hip prosthesis stem shows better long-term results than titanium. The latter bends more easily, which can lead to cement breakage and the formation of fragments, ultimately causing aseptic loosening [35].

According to the national joint replacement registries, polished, collarless and tapered stems with round edges and a rectangular cross-section provides good clinical outcomes [36,37].

Although today the implantation rate of cemented THA stems has decreased significantly compared to cemented stems, evidence shows better outcomes of cemented femoral components in elderly patients especially in females and those with overall poor bone quality [38].

CoCr proved to have the lowest level of roughness of the samples measured via the profilometer. Our findings are consistent with the data published by Revilla-Léon et al. [39]. Their study compared chemical composition (EDS), surface roughness (profilometry) and ceramic shear bond stress of milled and selective laser melted (SLE) CoCr surfaces. EDS and profilometry showed similar results when compared with our investigation.

Cobalt-based alloys are nonmagnetic, wear-, corrosion- and heat-resistant. The first medical application of cobalt-based alloys was in the cast of dental implants due to its excellent degradation resistance against the hostile oral environment. Currently, the medical applications of Co alloys are mostly for orthopedic prostheses of the hip, knee and shoulder and also in support of fracture fixation devices [40].

The main limitation of metal-on-polyethylene bearing was poor wear properties of UHMWPE, which resulted in the aseptic loosening of the components. Furthermore, the use of larger diameters further increased this risk. To solve this problem, the aim of the research in the 1980s and 1990s was to develop a more resistant bearing surface [41].

Retrospective clinical studies showed that the CoCr-HXLPE bearing type has low wear rates in THR, regardless of femoral head size [42].

CoCr alloys are the most frequently used materials for TKR femoral components. The procedure facilitates good postoperative joint function recovery and demonstrates excellent long-term follow-up results. Despite the good results, wear debris inducing osteolysis and aseptic loosening is still a concern and one of the major causes of TKR failure [43].

Its outstanding corrosion resistance is due to the thin Cr2O3 layer that forms on its surface during manufacturing, which reduces the outflow of metal ions into the tissues. According to studies, using additional manufacturing methods, the rate of ion outflow is even lower [44,45].

PE wear plays an important role in implant loosening after TKR. Civinini et al. compared the annual reports of the National Joint Registers and found higher 10-year cumulative revision rates for UHMWPE (5.8%) than HXLPE (3.5%) [46].

The investigated HXLPE and UHMWPE samples showed similar smooth and regular surfaces visualized by SEM. Their measured roughness was in the same range with a relatively low value with both AFM and profilometry. These results are consistent with its function as a bearing surface. EDS and XPS showed that the materials consisted mostly of carbon and oxygen as expected. Disparate O binding states on the investigated PE surfaces proved by XPS, are likely due to the applied radiation crosslinking method in manufacturing HXLPE material.

Particulate wear and delamination wear due to oxidation are historical issues associated with UHMWPE. As a solution to the problem, HXLPE created by irradiation was introduced to clinical practice. Better wear and oxidation resistance makes HXL a good option for THR, yet decreased mechanical properties when compared to UHMWPE are a cause of concern in its application in TKR. Another important phenomenon regarding HXLPE is the production of smaller wear debris particles which can be more biologically active [47].

Check et al. compared the friction behavior of two PE surfaces with different levels of roughness (Ra: 130 nm ± 19.9 vs. 62.5 nm ± 4.0) against the silicone–nitride interface in air and in the bovine serum. In a similar study, Gispert et al. investigated the wear characteristics of standard (Ra: 126 nm) and smooth (Ra: 22 nm) PE surfaces. They discovered, in the presence of protein, wear is very low, independent of its roughness [48,49]. In both studies, roughness measurements were carried out via AFM and one of the tested materials had a similar Ra value as our investigated PE surfaces (HXLPE: 147 ±13, UHMWPE: 144 ± 15).

## 5. Conclusions

Surface properties fundamentally determine the behavior of biomaterials used in orthopedics; therefore, they significantly influence long-term survival of the implanted devices. In recent decades, various processes have been developed to improve specific surface characteristics of the applied implants. Prior to the clinical application of these implants, thorough and accurate research is necessary to evaluate their different features. Appropriate methods can provide accurate information for physicians in selecting the most suitable device for the replacement of a specific function inside the human body. Chemical composition and roughness bear great impact upon the interactions between the applied biomaterial and its environment. Both biological and tribological properties directly affect their long-term performance. Our results indicate various prosthesis materials that have significantly different features according to their specific functions in musculoskeletal surgery. The detected high ratios of Ti^4+^ on the investigated Ti2 and TPS surfaces (93.9% and 81.3%, respectively) indicate good osseointegration ability according to the literature. A high average roughness value of TPS surfaces (R_a_: 25.5 ± 2.8 μm) is also a preferred feature for biological fixation, for example cementless THR stems and cups. Our measurements showed a low average roughness of the SS and CoCr implant materials, and the profilometry measured Ra value was 2.5 ± 0.1 μm and 1.0 ± 0.1 μm, respectively. Smooth surfaces are essential for the long-term cemented fixation of SS implants in THR and for low wear rates using CoCr-bearing surfaces in THR and TKR. Even surfaces of UHMWPE and HXL with the lowest roughness values (1.4 ± 0.1 μm and 1.5 ± 0.1 μm, respectively, via profilometry) of the examined materials can also be considered advantageous in terms of wear resistance regarding previous evidence. High resolution SEM images proved to be a useful method visualizing the measurements. Contact angle measurements and in vitro cell culture testing are further planned to test their surface energy characteristics and biocompatibility. Bacterial adhesion on the surface of implants is still an unsolved problem, so further microbiological studies are necessary to reduce the chance of periprosthetic joint infections (PJI).

## Figures and Tables

**Figure 1 polymers-16-01193-f001:**
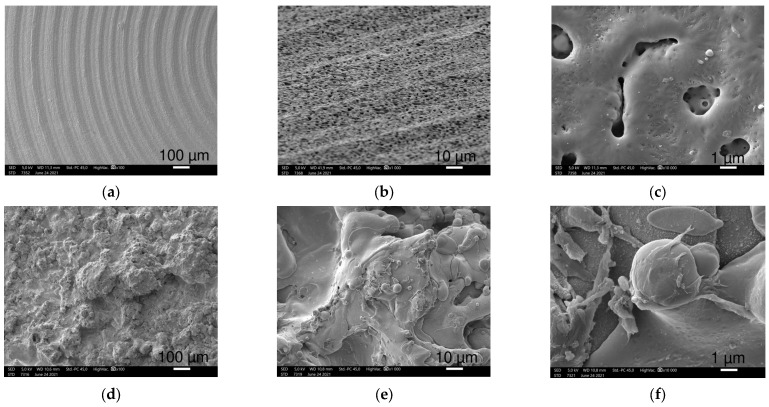
SEM images of Ti2 (**a**–**c**) and TPS (**d**–**f**) samples at 100×, 1000× and 10,000× magnifications (left to right), respectively. (**b**) was taken from a 45° tilted surface. Ti2 pictures demonstrate the grooved structure with countless openings, while pictures taken from TPS surfaces show rougher structures with round details.

**Figure 2 polymers-16-01193-f002:**
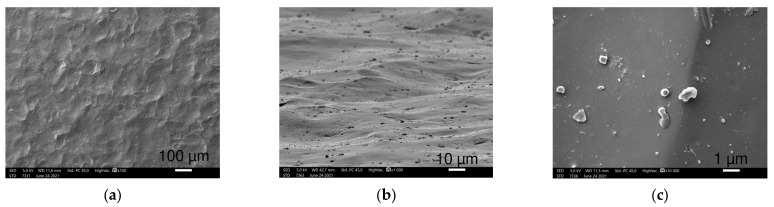
SEM images of SS (**a**–**c**) and CoCr (**d**–**f**) samples at 100×, 1000× (45° tilted) and 10,000× magnifications (**left** to **right**), respectively. SS shows wave-like pattern with small particles. CoCrMo surfaces shows a rougher surface with sharp-edged irregular regions.

**Figure 3 polymers-16-01193-f003:**
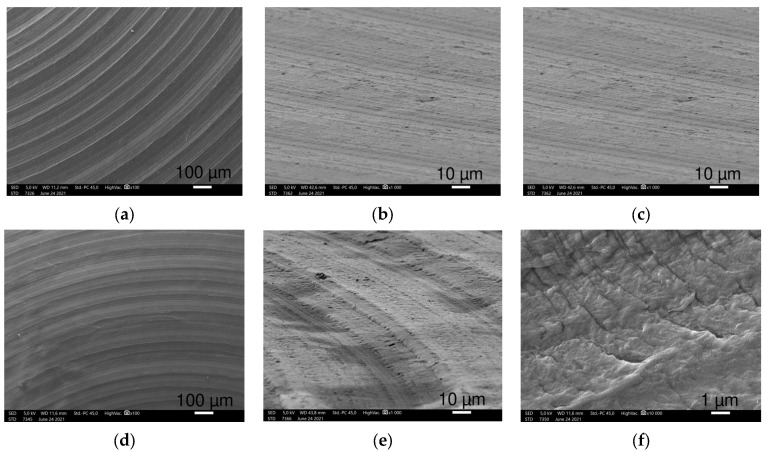
SEM images of HXL (**a**–**c**) and UHMWPE (**d**–**f**) samples at 100×, 1000× (45° tilted) and 10,000× magnifications (left to right), respectively. Both show circular creases at 100× and 1000× magnifications. Differences in patterns are visible at the highest magnification.

**Figure 4 polymers-16-01193-f004:**
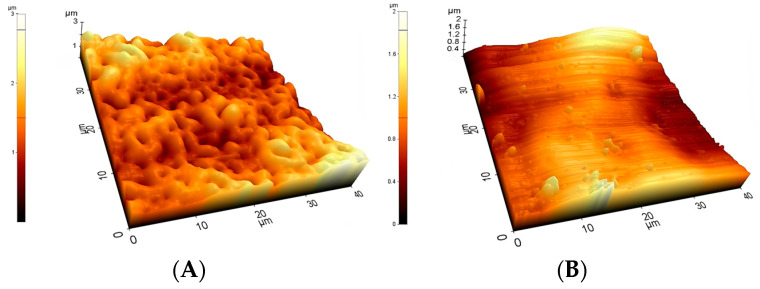
Typical 40 μm × 40 μm AFM topographies of the investigated materials. Upper row: Ti2 (**A**), SS (**B**), and CoCr (**C**). Lower row: HXL (**D**) and UHMWPE (**E**). Surfaces mapped by AFM showed similar morphologies when compared with the 100× magnification SEM images.

**Figure 5 polymers-16-01193-f005:**
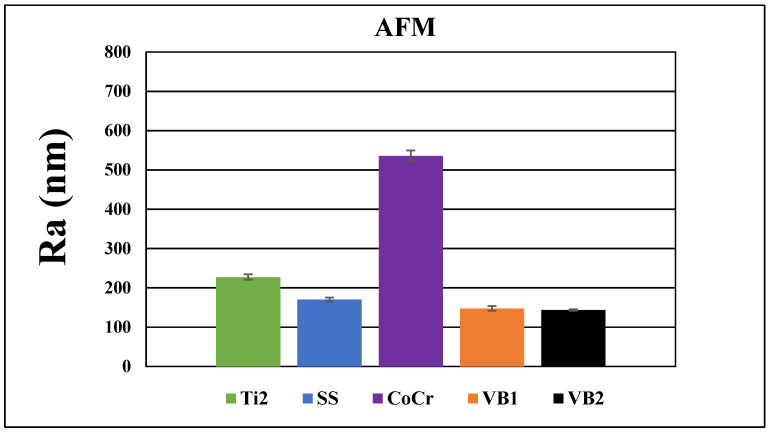
Average surface roughness and average ± SEM (standard error of the mean) values of Ti2, SS, CoCr, HXL and UHMWPE samples at 40 × 40 μm scan size.

**Figure 6 polymers-16-01193-f006:**
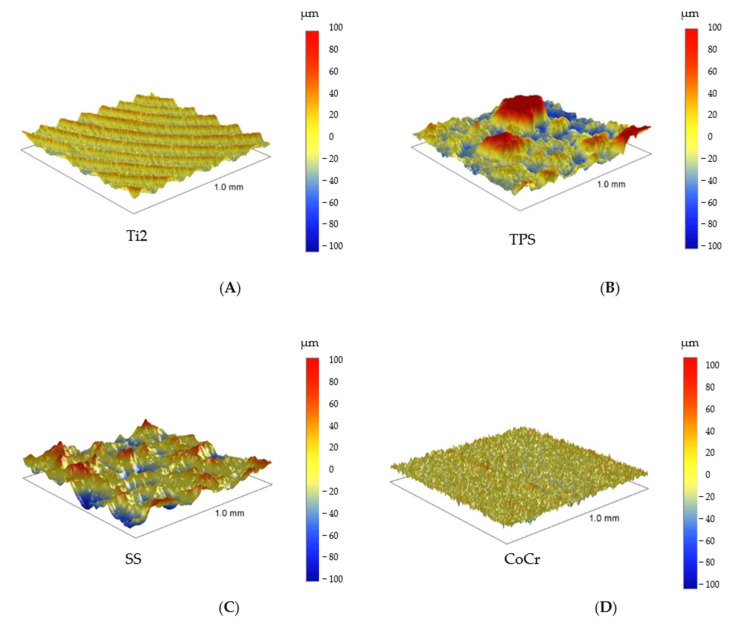
The 1000 μm × 1000 μm profilometry images of Ti2, TPS (upper row: **A**,**B**), and SS and CoCr (lower row: **C**,**D**).

**Figure 7 polymers-16-01193-f007:**
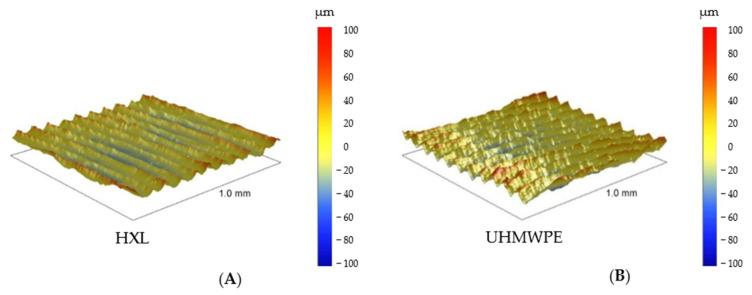
Profilometry topographies of HXL (left, **A**) and UHMWPE (right, **B**), 1000 μm × 1000 μm size images.

**Figure 8 polymers-16-01193-f008:**
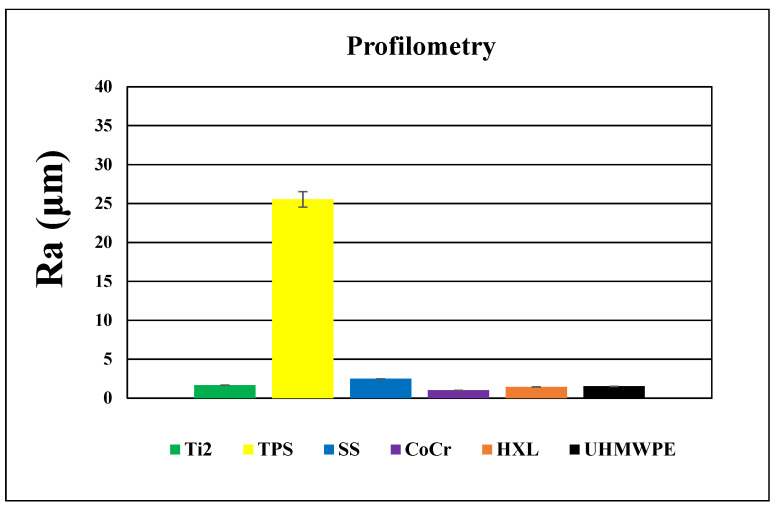
Average roughness and average ± SEM (standard error of the mean) values of the investigated materials measured by profilometry (1000 μm × 1000 μm size images).

**Figure 9 polymers-16-01193-f009:**
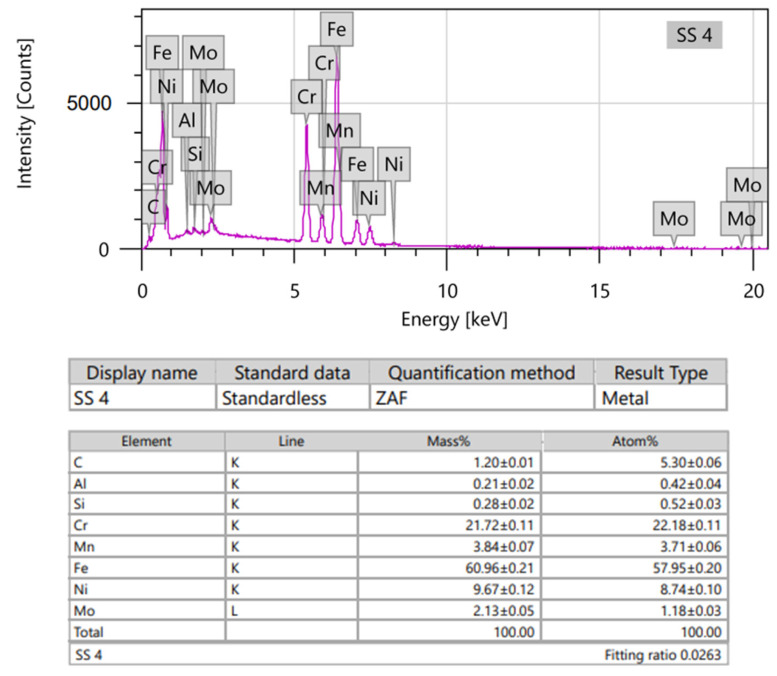
EDS spectra (**upper**) and element analysis (**lower**) of SS discs.

**Figure 10 polymers-16-01193-f010:**
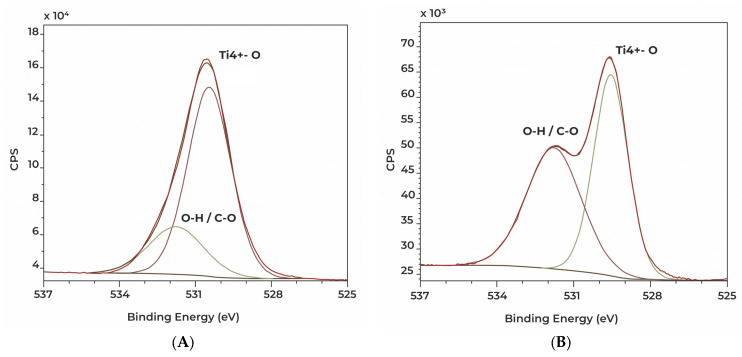
High resolution XPS spectra showing O1s signals of Ti2 (**A**) and TPS (**B**) discs.

**Table 1 polymers-16-01193-t001:** Atomic percentage of typical elements on the surface of Ti2 and TPS samples measured by EDS.

	Composition (in%)							
	O	Ti	C	P	N	Al	Si	Na
Ti2	69.16	22.44	7.87	0.51	0	0	0	0
TPS	29.04	42.81	4.12	0	12.04	11.51	0.35	0.13

**Table 2 polymers-16-01193-t002:** Atomic percentage of typical elements on the surface of SS and CoCr samples measured by EDS.

	Composition (in%)									
	Fe	Co	Cr	C	Al	Mo	Ni	Si	Mn	O
SS	56.5	0	21.6	6.9	0.53	1.38	9.09	0.53	3.52	0
CoCr	0	50.6	25	7.42	5.09	2.5	0	0.59	0	8.9

**Table 3 polymers-16-01193-t003:** Atomic percentage of typical elements on the surface of HXL and UHMWPE samples measured by EDS.

	Composition (in%)		
	C	Al	Ni
HXL	99.87	0.096	0.034
UHMWPE	99.61	0.39	0

**Table 4 polymers-16-01193-t004:** Atomic percentage of typical elements on the surface of Ti2 and TPS samples measured by XPS.

	Composition [in%]											
	C	O	Ti	Sn	P	Ca	Na	S	Zn	N	Si	Al
Ti2	44.6	**35**	**4.1**	0.8	10.7	0.9	0.1	0.4	0.1	2.7	0.6	-
TPS	56.6	**27.8**	**7.1**	0.4	-	1.7	0.1	0.6	-	0.9	4.4	0.5

**Table 5 polymers-16-01193-t005:** Ti binding state percentage on Ti2 and TPS samples measured by XPS.

	Ti Binding State (%)			
	Ti (IV)	Ti (III)	Ti (II)	Ti (Metal)
Ti2	**93.9**	2	4.1	-
TPS	**81.3**	8.2	7.1	3.7

**Table 6 polymers-16-01193-t006:** O binding state percentage on Ti2 and TPS samples measured by XPS.

	O Binding State (%)	
	Ti ^4+^–O	O–H/C = O
Ti2	**75.8**	**24.2**
TPS	**51.1**	**48.9**

**Table 7 polymers-16-01193-t007:** Atomic percentage of typical elements on the surface of HXLPE and UHMWPE samples measured by XPS.

	Composition [in%]								
	C	O	N	Cl	Ca	Si	S	Zn	Na
HXLPE	87.7	8	1.7	0.7	0.6	1.1	0.2	-	-
UHMWPE	90.3	6.4	0.2	0.8	1.1	0.8	0.2	0.1	0.1

**Table 8 polymers-16-01193-t008:** O binding state percentage on HXLPE and UHMWPE samples measured by XPS.

	O Binding State (%)	
	O=C, O-H	O-C
HXLPE	**76**	24
UHMWPE	**100**	-

## Data Availability

Data are contained within the article.

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
