# Peer review of "Characterizing Surface Morphological and Chemical Properties of Commonly Used Orthopedic Implant Materials and Determining Their Clinical Significance"

_polymers, 2024, doi:10.3390/polym16091193_

Round 1

Reviewer 1 Report

Comments and Suggestions for Authors

After thoroughly evaluating the manuscript titled "Surface Morphological and Chemical Properties of Commonly Used Orthopedic Implant Materials," it is evident that the study significantly advances our comprehension of orthopedic implant materials, offering valuable insights for clinicians, researchers, and manufacturers in orthopedic surgery and traumatology. However, the manuscript exhibits a considerable degree of dismissal in its wording, leading to a 36% match in the iThenticate report. To enhance clarity and originality, several revisions and clarifications are necessary.

The abstract should be refined to succinctly summarize the key findings and implications of the study without unnecessary repetition.

The introduction needs to be revised to highlight the novelty of the study and provide a clear rationale for the research. Incorporating recent literature to demonstrate the current state of knowledge in the field and the gaps that the study aims to address will enhance the introduction's effectiveness

The conclusion should succinctly summarize the main findings of the study and their implications for clinical practice, research, and manufacturing. It should also provide recommendations for future research directions based on the study's outcomes.

Bar scale in SEM image is not clear.

Can you explain the significance of comparing the morphological and chemical structure of different implant materials in orthopedic surgery and traumatology?

What were the key findings regarding the elemental composition of the investigated prosthesis materials as revealed by energy dispersive spectroscopy (EDS)?

In what ways did the X-ray photoelectron spectroscopy (XPS) results complement the findings from EDS analysis?

How do the observed surface features and characteristics of the implant materials influence their suitability for different orthopedic procedures?

What implications do the differences in surface properties of the implant materials have for their clinical performance and long-term outcomes?

Can you discuss the potential implications of the high percentage of Ti4+ observed on the Ti2 and TPS surfaces in terms of biocompatibility and osseointegration?

Based on the study findings, what recommendations would you propose for optimizing the selection and application of orthopedic implant materials in clinical practice?

Addressing these issues will strengthen the manuscript's impact and readability, ensuring its relevance to a broad audience of stakeholders in orthopedic surgery and traumatology.

Author Response

Thank you very much for taking the time to review this manuscript. Please find the detailed responses below and the corrections highlighted in the re-submitted files.
Please see the attachment.

Reviewer 2 Report

Comments and Suggestions for Authors

·        Paper titled “Surface Morphological and Chemical Properties of Commonly 2 Used Orthopedic Implant Materials” is recommended for publication after revision.

·        Change the title- the present one looks like review paper title

·        In Figure 1. SEM images of Ti2 (A, B, C) and TPS (D, E, F) samples – provide the scale and scale value legibly.

·        Figure 2. SEM images of SS (A, B, C) and CoCr (D, E, F) samples - provide the scale and scale value legibly.

·        Figure 3. SEM images of HXL (A, B, C) and UHMWPE (D, E, F) samples - provide the scale and scale value legibly.

·        We performed AFM examinations on the Ti2, SS, CoCr, HXL and UHMWPE samples – improve the font size

·        Figure 4. Typical 40  m x 40  m AFM topographies of the investigated materials - improve the font size

·        Figure 5. Average surface roughness – improve the quality of the image

·        Due to its high mechanical properties and wear resistance, UHMWPE is the most 487 widely used bearing material in joint arthroplasties since its introduction in 1962 – this should be in introduction section. Why in the discussion section?

·        Address the novelty of the work in the introduction section

·        Revise the conclusion section with outcome and not too general

·        Why two times the references numbers are given in the reference section

·        Investigations were performed on the effect of roughness on the tribological behavior 495 of UHMWPE. – this should be in experimental section

·        Cobalt based alloys are nonmagnetic, wear-, corrosion and heat-resistant – have authors presented any corrosion studies here?

Author Response

(The authors gave the same response as above.)

Reviewer 3 Report

Comments and Suggestions for Authors

The authors of the manuscript carried out investigations on the topic “Surface Morphological and Chemical Properties of Commonly Used Orthopedic Implant Materials”. The problem of biointegration of materials used in medicine is largely determined by the condition and quality of their surface. For example, titanium ions and nanoparticles as degradation of medical products under conditions of functional load can lead to mucositis and peri-implantitis. Therefore, the research carried out in this work are important, relevant and of scientific interest.

There are several comments:

1.Write a conclusion more specifically, include in the text quantitative characteristics of the surfaces of the materials studied.

2.Improve quality Fig. 9, 10. Small letters and symbols are difficult to see.

 3.Add DOI to the reference list for each article in accordance with the journal’s rules.

Author Response

(The authors gave the same response as above.)

Round 2

Reviewer 1 Report

Comments and Suggestions for Authors

The revised version is recommended for publication..